# Lightweight Strong PUF for Resource-Constrained Devices

Mateusz Korona ⬡, Radosław Giermakowski ⬡, Mateusz Biernacki ⬡ and Mariusz Rawski *⬡

Institute of Telecommunications, Faculty of Electronics and Information Technology, Warsaw University of Technology, 00-665 Warsaw, Poland; mateusz.korona@pw.edu.pl (M.K.); radoslaw.giermakowski@pw.edu.pl (R.G.); mateusz.biernacki@pw.edu.pl (M.B.)
* Correspondence: mariusz.rawski@pw.edu.pl

**Abstract:** Physical Unclonable Functions are security primitives that exploit the variation in integrated circuits' manufacturing process, and, as a result, each instance processes applied stimuli differently. This feature can be used to provide a unique fingerprint of the electronic device, or as an interesting alternative to classic key storage methods. Due to their nature, they are often considered an element of the Internet of Things nodes. However, their application heavily depends on resource consumption. Lightweight architectures are proposed in the literature but are technology-dependent or still introduce significant hardware overhead. This paper presents a lightweight, Strong PUF based on ring oscillator architecture, which offers small hardware overhead and sufficient security levels for resource-constrained Internet of Things devices. The PUF design utilizes a Linear Feedback Shift Register-based scramble module to generate many challenge–response pairs from a small number of ring oscillators and a control module to manage the response generation process. The proposed PUF can be used as a Weak PUF for key generation or a Strong PUF for device authentication.

**Keywords:** Physical Unclonable Functions; lightweight; Internet of Things; cybersecurity

## 1. Introduction

The Internet of Things (IoT) has become a reality with the increasing emergence of mobile electronic devices over the last two decades. Its influence on our day-to-day activities will increase and more than 30 billion devices are expected to be connected worldwide by 2025 [1]. The Internet of Things is a set of connected devices, each with a unique identifier that automatically collects, processes, and exchanges data over a network. Due to the unconventional manufacturing of IoT devices and the vast amount of data they handle, there is a constant threat of cyber attacks. Unfortunately, remarkably inexpensive and computationally limited IoT devices are generally not designed with security mechanisms. In addition, IoT device manufacturers may also compromise on security measures to keep pace with market needs, for example, by reducing the time to market for their devices and minimizing the design and development costs.

System security in IoT devices for applications requiring user identification and authentication is often based on protecting secret keys. Malicious users can impersonate authorized users in these applications, when the secret key is uncovered. To protect integrated circuits (ICs), that are the base of IoT systems, against logical and physical tampering attacks, secret key storage devices provide active logical controls to safeguard the secret keys. However, an adversary can extract digitalized secret information from the device by applying invasive and non-invasive physical tampering methods such as micro-probing, laser cutting, glitch attacks, or power analysis. For instance, an attacker can apply sophisticated reverse engineering to build copies of the circuits and compromise conditional access systems by using illegal copies of secret information. The device authentication method provided in most IoT platforms relies on storing the key in memory. This poses a security threat due to the exposure of the authentication key, as even data in non-volatile memories, such as EEPROM and NVRAM, can be exposed using sophisticated manipulation methods.

This has led to a high demand for cryptographic mechanisms to protect user privacy and data security.

The notion of a Physical Unclonable Function (PUF) has been introduced in [2]. PUFs are security primitives that utilize the inherent, uncontrollable process variation during integrated circuit manufacture to generate a unique intrinsic identifier for a device. This physical entity, when stimulated by the input value (challenge), by exploiting process variations and mismatch, produces a hard-to-clone output (response) that can be used as a cryptographic key or unique fingerprint for (the physical instance of) a device. When PUF is utilized, unique keys are not stored physically on the device (unlike in classical key storage methods such as non-volatile EEPROM / NVRAM memories), but are rather (re-)generated by PUF upon each activation. Additionally, reverse engineering of PUF-based devices can be challenging due to the unpredictability of manufacturing process variations. This particular feature has given birth to new PUF-related security applications. A comprehensive review of the state of the art of PUF, its architectures, protocols, and security for IoT has been provided in [3].

PUFs can be implemented in Application-Specific Integrated Circuits (ASICs), or, at a lower cost, in Field-Programmable Gate Arrays (FPGAs). FPGA devices are popular for IoT and embedded systems applications due to their programmability, partial reconfigurability, and faster time-to-market features. Many PUF architectures have been proposed for FPGA. A ring oscillator PUF (RO PUF), that has been introduced in [4], is an FPGA-friendly design using the delay of FPGA primitives. Various industrial and academic efforts have attracted the development of different techniques to improve the performance of PUFs using FPGAs to find a reliable and robust countermeasure against increased hardware security attacks [5–7].

In general, PUFs can be classified into two subtypes—Weak PUFs and Strong PUFs [8]. The difference between the two is the number of possible challenge–response pairs (CRPs) supported by the PUF. If the number of CRPs supported by the PUF scales exponentially with its size, it is considered a Strong PUF. On the other hand, linear or polynomial increases typically characterize Weak PUFs. Exponential scaling produces exceptionally large CRP sets with increasing device size. Strong PUFs can be used directly for authentication, while Weak PUFs are used for cryptographic key generation [2].

A Strong PUF realization faces some challenges regarding resource utilization and security level. Secure PUF may require a large amount of resources to implement [9,10], while simplified PUF implementation may be vulnerable to machine learning attacks that can model its behavior [11]. The security concerns for IoT center around resource-constrained IoT devices with limited processing power and storage capabilities. In designing cryptographic primitive for such devices, there is always a tradeoff between the security level provided by the primitive and the resources needed to implement it. Therefore, designing a PUF that balances resource utilization and security level is an important and active research topic in the field of hardware security.

In [12], a lightweight Strong PUF design has been proposed. This Arbiter-based PUF (APUF) is composed of two identical groups of flip-flops and multiplexers. The response generation is dependent on creating a race condition between two identical delay paths. Although it saves more than 66% in hardware resource usage compared to previous APUF designs [13], its hardware overhead is still significant.

In [14], a p-SPUF ("pseudo" Strong PUF) is proposed that combines Weak PUF with obfuscation logic in the form of shift registers. Although its simple architecture, it delivers good resistance to modeling attacks and smaller hardware overhead than typical Strong PUF designs. However the proposed solution is not technology independent since it uses some structural features of selected Xilinx FPGA.

Another example of a lightweight solution is an LFSR-based Strong PUF introduced in [15]. Linear Feedback Shift Register (LFSR), as a circuit that can generate pseudorandom number sequences, is widely used in key generation and communication fields. LFSR is usually composed of two parts: a shift register and a feedback function (polynomial). The

Strong PUF structure proposed in [15] uses the output response of the Weak PUF to provide a unique feedback polynomial for the LFSR, making the LFSR structure of each device different. However, since the response of a given PUF depends on uncontrollable process variation during integrated circuit manufacturing, the resultant feedback function may lead to the suboptimal polynomial. If the polynomial is not a primitive one, the generated sequences may be easily predictable. This makes the sequences obtained from such an LFSR not directly usable in security applications [16]. Moreover, as the authors admit in their paper, using LFSR in the described manner will cause reliability problems because it will amplify the unreliability of utilized Weak PUF. To mitigate this problem, additional resource-intensive reconstruction mechanisms would be required.

This paper proposes a lightweight Strong PUF based on ring oscillator architecture. The presented approach utilizes the LFSR-based scrambler to control a set of ring oscillators to sequentially generate bits of the PUF response. It was implemented on several FPGA-based platforms and experiments were conducted to verify its efficiency. Obtained results demonstrate that such realization offers small hardware overhead. Significantly reduced complexity results in low resource utilization in comparison to the existing solutions. It delivers substantial security with a high uniqueness and reliability level for resource-constrained IoT devices. The proposed approach is not technology dependent and can be used to implement PUFs in ASIC, as well as in FPGAs.

## 2. Basic Information

### 2.1. Physical Unclonable Function

A PUF is a physical system that can be challenged with external stimuli or so-called "challenges" $C_i$. Depending on the PUF type, it can have one possible challenge, a few, or even a large number. When exposed to a challenge $C_i$, the PUF reacts by producing a corresponding response $R_i$. The tuples $(C_i, R_i)$ are termed the challenge–response pairs (CRPs) of the PUF. PUF can produce unique $n$-bit responses with an $m$-bit challenge, depending on the process variation during integrated circuit manufacture (Figure 1).

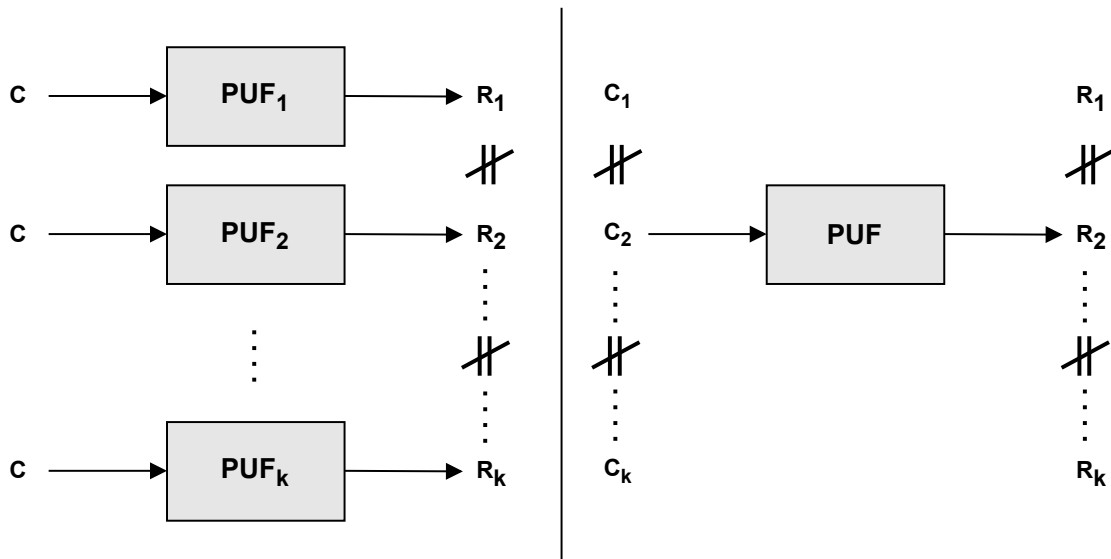

**Figure 1.** Working principle of PUF module.

### 2.2. Ring Oscillator PUF

The Ring Oscillator PUF is a popular FPGA-based PUF topology. The basic PUF implementation technique based on ring oscillators uses the architecture shown in Figure 2, proposed in [4]. It consists of $N$ Ring Oscillators (RO), which oscillate with a particular frequency. Despite being designed identically with an odd number of inverters, each ring oscillator (RO) has a different output frequency due to process variations in each inverter

stage. To generate a single response bit, a specific pair of oscillators is selected by an input called "challenge". The challenge is passed as an select input to the multiplexers (*ctrl_1*, *ctrl_2*) in order to set the corresponding oscillators to their outputs. The output of each multiplexer drives the clock input of a counter that measures the frequency of the selected oscillator, which is enabled for a fixed period. Then, depending on which counter holds the higher value, a response bit of 1 or 0 is generated. To generate more bits, the procedure is repeated with different challenge values.

In [3], the authors describe several RO-PUF solutions suitable for IoT applications and discuss the strengths, weaknesses, quality metrics, and evaluation of common architectures.

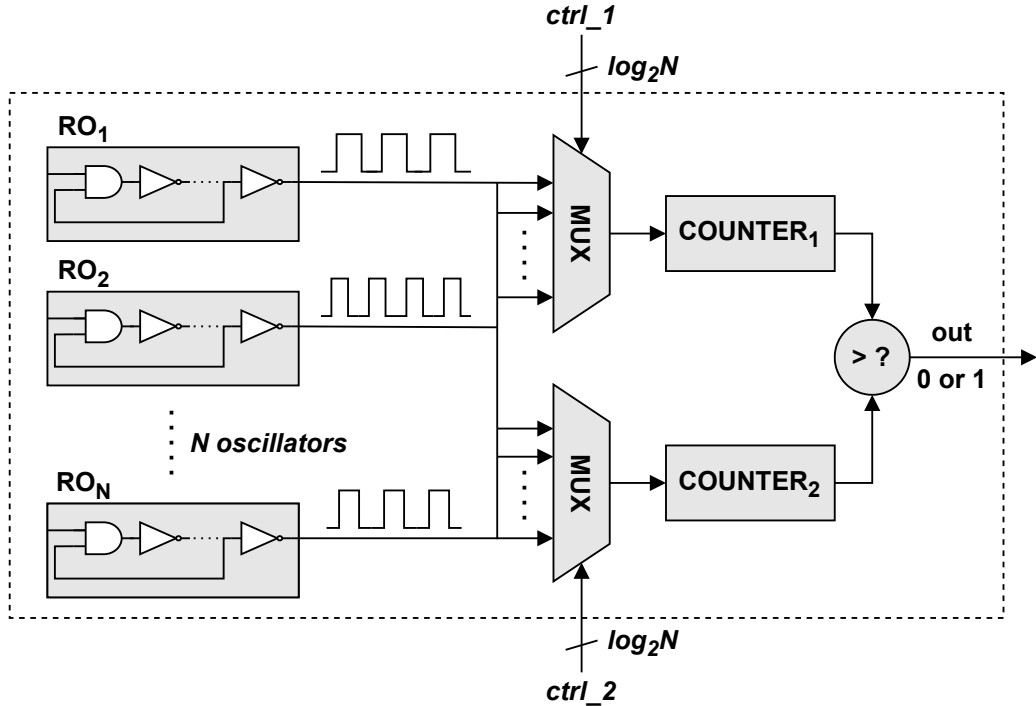

**Figure 2.** The traditional structure of the RO PUF. Based on [4].

## 3. Lightweight RO PUF Concept

The basic RO PUF circuit shown in Figure 2 can generate $N(N-1)/2$ response bits, where $N$ is the number of ring oscillators. The bit width of the PUF response equals to the number of combinations of oscillator pairs whose frequencies are compared. However, the number of independent bits that can be generated is less than $N(N-1)/2$ because the bits obtained from the frequency comparisons of the ring oscillators already used may correlate. Each oscillator should be used only once to generate a single bit to avoid this situation. However, this involves an increase in system resource consumption since generating, for example, 128 response bits requires the implementation of as many as 128 distinct pairs of ring oscillators. The described solution is characterized by high sensitivity to temperature changes and high power consumption, but good statistical properties of the generated response.

To increase the number of response bits produced by the PUF, the concept of duplicating one-bit response PUF instances can be used [17]. It assumes multiple, parallel-connected RO PUF submodules, whose number depends on the number of expected bits in a response. The application of such a concept is characterized by a constant response generation time equal to the number of clock ticks needed to generate a single bit, where the generated response bits are not correlated. In addition, the result obtained is characterized by good statistical properties, i.e., high uniqueness between different devices and a uniform distribution of zeros and ones. The main disadvantage of the described solution is the significant resource consumption. Assuming that a single RO PUF submodule consists of 32 ring

oscillators, the generation of 16 response bits requires 512 oscillators and an appropriately adjusted number of multiplexers or counters, of which a single RO PUF instance consists. The linear dependence of the number of ring oscillators used on the number of response bits makes this type of architecture unacceptable for implementing in IoT devices when creating cryptographic keys of the lengths used in typical cryptographic functions (e.g., 128 bits in AES).

To apply RO PUFs in resource-constrained IoT devices, a compromise has to be found between the statistical properties of the generated response and the hardware resources required for implementation. The proposed solution is based on the serial architecture of RO PUF that has been enhanced with an additional module generating a sequence of challenges to produce a response of a required number of bits (Figure 3). This allows us to reduce the number of ring oscillators used and, thus, a significant reduction in the number of circuit's logic elements is achieved. The sequence of challenges is generated by a scrambler based on some initial value. Each successive response bit is generated based on a different challenge—the subsequent value in the scrambler's stream.

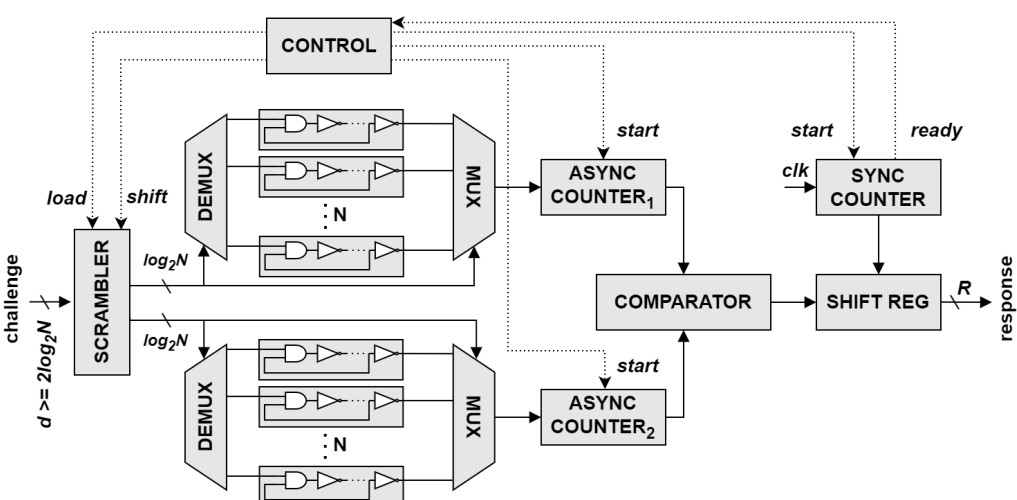

**Figure 3.** The lightweight RO PUF concept.

The scrambler's most straightforward implementation may be using a simple counter that increments or decrements the challenge value. However, this leads to the creation of correlated output values due to the linearity of the counter. To avoid a high correlation of the response bits and thus increase the level of uniqueness of the responses for different challenges, a scrambler structure should be based on a pseudo random generator. The most common method to implement a pseudo-random number generator is using a Linear Feedback Shift Register (LFSR). This method is preferable due to it's low-cost implementation, large period, and good statistical properties [18]. However, this will also lead to a correlation of some of the response bits, for given pairs of challenges (initial values of the LFSR), if they are incorrectly selected. This is due to the periodicity of the LFSR—for example, if eight response bits are generated and values $X$ and $Y$ are adjacent in the LFSR cycle, the response to challenge $X$ and challenge $Y$ will have seven bits in common, leading to a reduction in intra-chip uniqueness.

To solve this problem, some degree of non-linearity needs to be introduced, which can be realized by calculating a simple *XOR* of the values from the LFSR with the initial challenge value. This solution will allow for the generation of two different challenge sequences for both initializing values $X$ and $Y$, and increasing the uniqueness value. Subsequent values from the LFSR, on the other hand, are calculated from a generating polynomial. Appropriate taps (representing the primitive polynomial) for maximum length LFSR have been selected from [19]. The values generated by the scrambler are used to control two demultiplexers and two multiplexers, each having $N$ ring oscillators connected to them so that a specific pair of oscillators is selected whose frequencies will be compared for each

response bit. The function of the demultiplexers is to appropriately control the enable signal so that only one RO is triggered at a time, limiting the system's power consumption. The bit width of the values generated by the scrambler module should correspond to the number $2log_2N$, where $N$ is the number of ring oscillators used for a single multiplexer. The number of unique challenge values is equal to $2^d - 1$, where $d$ specifies the number of bits of the scrambler. The number of CRPs scales exponentially, which is characteristic of Strong PUFs.

The CONTROL module (Figure 2) is the component that manages the operation of the PUF and the generation of subsequent response bits. It controls two asynchronous counters, a synchronous counter, and a scrambler. According to the pseudocode presented in Algorithm 1, the generation process begins in the *LOAD_CHALLENGE* by asserting *load* signal. This enables the loading of a new *challenge* value into the SCRAMBLER. Assigning an initial value to the SCRAMBLER is a one-time action during the generation of a single PUF response. Then, in the *SHIFT_SCRAMBLER* state, the *shift* signal is asserted to enable the calculation of the MUX/DEMUX selection value. Subsequently, the corresponding pair of ring oscillators is activated, and the rising edges of the signals they produce are counted by ASYNC COUNTERS. This process is initiated by a *start* signal (*START_COUNTERS* state). Then, on the expiry of a fixed time, indicated by the SYNC COUNTER (*ready* signal), a value is read from the COMPARATOR module, which determines the value of a particular bit of the response. The response register is implemented as a serial-in parallel-out shift register. Afterwards, the SCRAMBLER content value is modified, and the process is repeated until $R$ output bits have been generated. The proposed concept does not assume using a specific number of inverters comprising a single RO. This value can be set in such a way as to obtain the appropriate statistical properties of the generated response.

---

**Algorithm 1:** Generating PUF response

---

**Input:** start, load, shift, ready, comparator
**Output:** response

1 **Constants:** LOAD_CHALLENGE, SHIFT_SCRAMBLER, START_COUNTERS, WAIT_FOR_RESULT, R
2 **Variables:** current_state $\leftarrow$ IDLE , iterator $\leftarrow$ 0

3 **while** *iterator $<$ R* **do**
4     **if** *current_state == LOAD_CHALLENGE* **then**
5         start $\leftarrow$ false
6         load $\leftarrow$ true
7         shift $\leftarrow$ false
8         current_state $\leftarrow$ SHIFT_SCRAMBLER
9     **else if** *current_state == SHIFT_SCRAMBLER* **then**
10         load $\leftarrow$ false
11         shift $\leftarrow$ true
12         current_state $\leftarrow$ START_COUNTERS
13     **else if** *current_state == START_COUNTERS* **then**
14         shift $\leftarrow$ false
15         start $\leftarrow$ true
16         current_state $\leftarrow$ WAIT_FOR_RESULT
17     **else if** *current_state == WAIT_FOR_RESULT* **then**
18         **if** *ready* **then**
19             iterator++
20             response $\leftarrow$ { response[$R - 2 : 0$], comparator }
21             current_state $\leftarrow$ SHIFT_SCRAMBLER
22 **end**

---

If the scrambler's initial value (challenge) is loaded to the PUF every time at the beginning of the response generation, the proposed solution can be used as Strong PUF as discussed in previous paragraphs. The proposed architecture allows for the construction of Weak PUF as well, when the initial value (challenge) of the scrambler is fixed at the design time, and thus, only single CRP is available. In such case, the configuration and control logic can be further reduced.

### 4. Implementation

The proposed RO PUF architecture has been implemented on Cora Z7-10 evaluation boards with Zynq 7000 devices equipped with 28 nm programmable logic. Xilinx FPGA devices are built from configurable logic blocks (CLB), which can implement sequential or combinational functions. For the Zynq 7000 programmable logic, each CLB contains two blocks called SLICEs, each composed of four six-element display boards (LUT6), eight memory elements, a wide-function multiplexer, and carry logic [20]. To ensure the identical structure of each oscillator, it is necessary to use predefined macros provided by the CAD tool to define each single inverter or NAND gate in a single LUT6 instance. This is achieved by the correct description of the truth table of each LUT6 instance. Additionally, it was necessary to use the *dont_touch* macro, thus preventing the optimization of the implemented logic. To implement the feedback and output to the multiplexer, it is required to use the LUT6_2 primitive, which has two independent outputs implemented as two separate LUT tables. An example of mapping a ring oscillator consisting of four NOT gates, a NAND gate, and an additional LUT6 table implementing the feedback in the physical resources of the FPGA, which is shown in Figure 4.

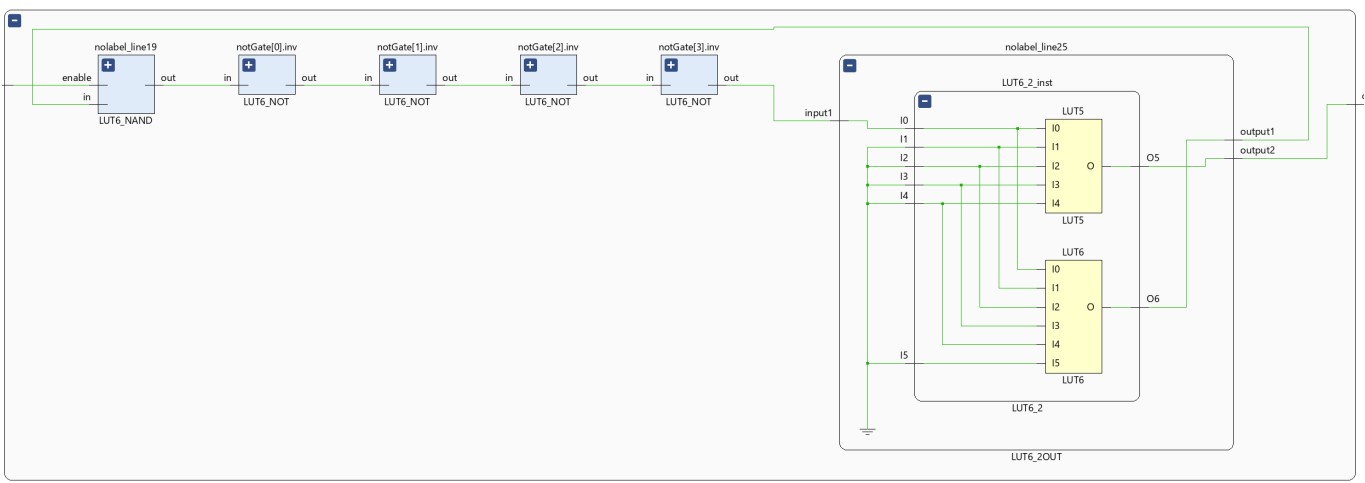

**Figure 4.** RO implementation in Zynq 7000 programmable logic.

To guarantee the randomness of each ring oscillators' pair comparison, it is necessary to ensure identical implementation of each RO in the FPGA's resources to avoid the impact of placement and routing procedure. The optimal solution is to place each oscillator in one single CLB block. However, this depends on the number of inverters used in a single RO. Unfortunately, the CAD tool randomly places the defined LUT6 instances on the available FPGA logic resources. Therefore, it is necessary to precisely specify each LUT instance's location, assign it to an appropriate SLICE block, and then to a specific CLB block. To do this, the use of *set_property BEL* and *set_property LOC SLICE* macros is necessary, which allows us to control the technological mapping process and ensures the identical structure of each ring oscillator. An example implementation of 64 ring oscillators for the Cora Z7-10 platform is shown in Figure 5. Each RO comprises six inverters, a NAND gate, and the LUT implementing feedback.

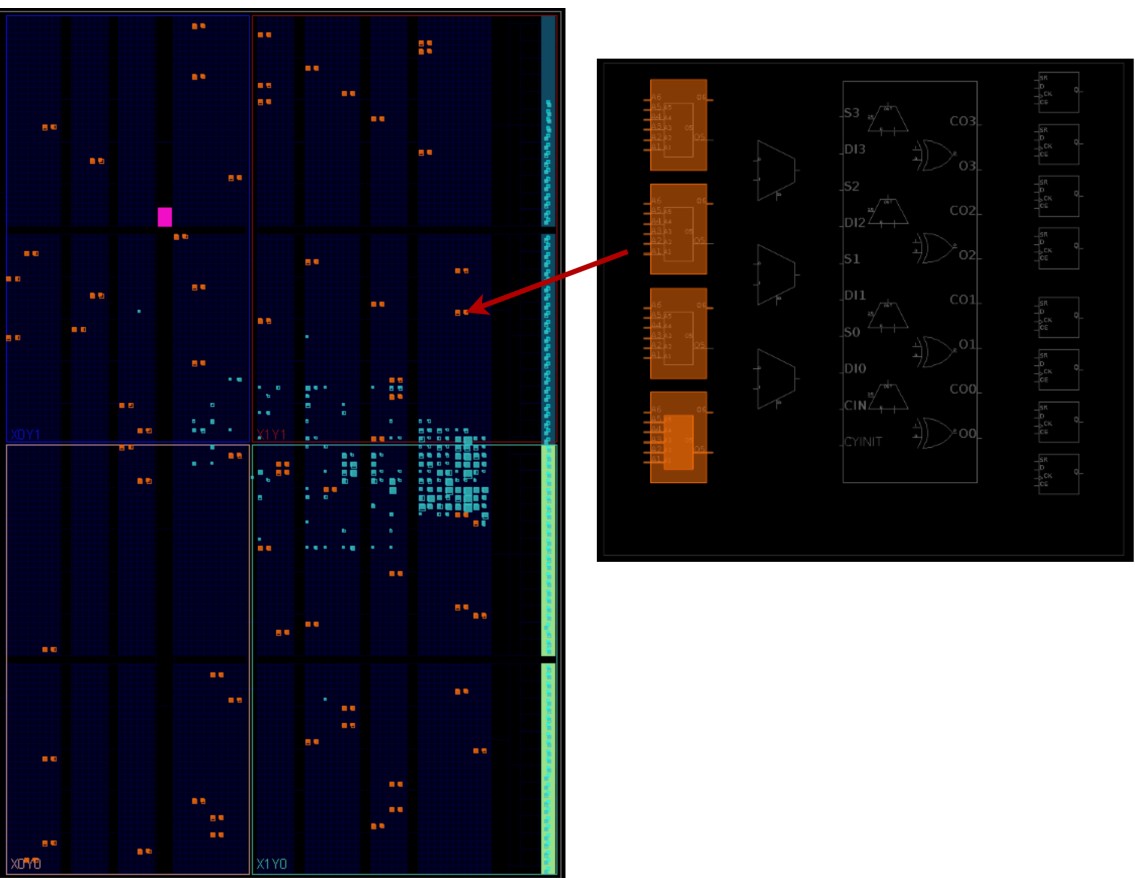

**Figure 5.** Implementation of ring oscillators in FPGA resources.

## 5. Experimental Results

Four parameters are used to evaluate the implemented RO PUF quality [21]. The first parameter is uniformity, which determines how uniform the distribution of "0" and "1" in the PUF response bits is. In ideal conditions, this proportion should be 50%. The next parameter is intra-chip uniqueness, which determines the degree of uniqueness of the PUF response on a single device but for different challenge values. Another key aspect is to consider inter-chip uniqueness, which defines the diversity degree of generated responses for the same challenge value but on different devices. An optimal value for uniqueness is 50%. Whether generating identifiers across different devices or cryptographic keys on a single device, it is crucial that the values obtained from the PUF differ significantly. The final parameter is reliability, which determines the degree of repeatability of generated responses for the same challenge value. The ideal reliability value should be equal to 100%, as it is essential to consistently reproduce the generated response.

Due to the limited number of hardware resources available in IoT devices, which are the target of the implemented PUF module, only ring oscillators consisting of three, five, and seven inverters were considered in the analysis. A PUF architecture consisting of 64 ring oscillators, 10-bit scrambler (based on maximum length LFSR with characteristic polynomial: $x^{10} + x^7 + 1$ [19]), 16-bit asynchronous counters, and an 8-bit reference counter was chosen for testing purposes. The system generated a 64-bit response. This size of the response is motivated by the size of the key for typical lightweight cryptographic algorithms, which is usually a multiple of 64 bits. This value is also selected in other lightweight PUF implementation.

Table 1 presents the results showing the parameters' dependence on the number of inverters used in a single RO. The tests were performed on four different Digilent Cora Z7-10 platforms. The inter-chip uniqueness test involved generating PUF responses for 1024 challenges on four devices. Conversely, the intra-chip uniqueness test calculated

Hamming distances of PUF responses for 1024 challenges on one device. All challenge values were randomly generated.

**Table 1.** Quality comparison for different RO lengths.

| RO Inverters | Uniformity [%] | Uniqueness (Inter-Chip) [%] | Uniqueness (Intra-Chip) [%] | Reliability [%] |
|:---:|:---:|:---:|:---:|:---:|
| 3 | 48.239 | 37.018 | 49.612 | 96.685 |
| 5 | 50.661 | 30.612 | 49.889 | 97.773 |
| 7 | 51.509 | 32.612 | 49.302 | 97.826 |

The distribution of uniqueness of PUF responses implemented in four different FPGA devices for these 1024 challenge values has been presented in Figure 6. According to the histogram and Table 1, the average inter-chip uniqueness is 37.018%. This value, satisfactory for most PUF applications [22], can be controlled by adding more RO at the cost of increased resource consumption. The histogram's shape approximates a normal distribution with a high standard deviation of 9%. Crucially, the quality of the generated responses strongly depends on the challenge value used to initialize the scrambler module. Thus, it is possible to define a range of challenges that would yield uniqueness close to the ideal value of 50%. In this study, a 10-bit scrambler was used, and all available initial challenge values (1024) were parsed. In the literature, the number of challenges used to calculate uniqueness is usually not defined, making it difficult to replicate results that are close to 50%.

The presented solution can be scaled in two ways. One way is to increase the size of the scrambler preserving the number of ROs, which will increase the number of CRPs but may reduce the uniqueness value. This solution will not affect hardware overhead much. The other way is to simultaneously increase the number of ROs, which will enhance both, the number and the quality of CRPs, at the expense of greater use of device resources.

Table 2 presents the resource utilization for RO PUFs dependent on the number of inverters. The results indicate that the proposed architecture of the PUF module is designed with low demand for logic elements of the system, corresponding to its use in resource-constrained devices.

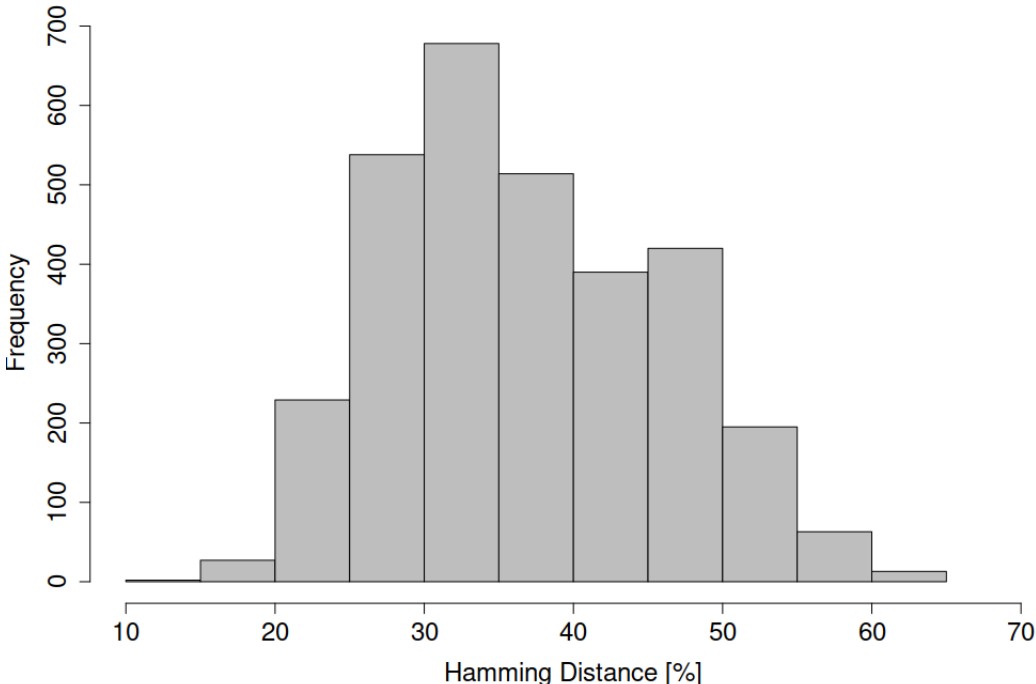

**Figure 6.** Distribution of fractional uniqueness values of PUFs between devices.

**Table 2.** Resource utilization.

| RO Inverters | Slice LUTs | Slice Registers | Slices |
|:---:|:---:|:---:|:---:|
| 3 | 474 | 148 | 174 |
| 5 | 603 | 148 | 232 |
| 7 | 730 | 148 | 235 |

In Table 3, a comparison between the presented PUF solution with three inverters in a single ring oscillator and various implementations of other PUFs is presented. Both the statistical values of the generated responses and the utilization of resources were compared. The proposed architecture results in resource consumption significantly lower than the Strong PUF solution based on arbiters. It also offers lower hardware overhead than Strong PUFs presented in the literature as lightweight. In fact, the resource consumption of the PUF solution proposed in this paper is at the level offered by Weak PUFs. This developed version of the PUF can generate a larger number of challenge–response pairs compared to the Weak PUF, using only 174 Slices cells while maintaining a high reliability value. Despite its lower uniqueness value, the proposed solution proves competitive with other PUF versions as a lightweight alternative.

**Table 3.** Comparison of hardware resource consumption and metrics of different PUF designs, based on [6]. Strong PUFs marked with * are presented in literature as lightweight. Results marked with † were obtained with small number of challenges. Results marked with ‡ present in fact reliability of Weak PUF used in the solution, but not the solution itself.

| PUF Design | Type | Uniqueness | Reliability | Hardware | Response (bit) | Resource Consumption |
|:---:|:---:|:---:|:---:|:---:|:---:|:---:|
| Latch PUF [23] | Weak | 46% | >87% | Spartan 3 | 128 | $2 \times 128$ slices |
| Flip-flop PUF [24] | Weak | ≈50% | >95% | Virtex 2 | 4096 | 4096 flip flops |
| Buskeeper PUF [25] | Weak | ≈50% | 94% | Virtex 5 | 64 | 130 slices |
| RO PUF [4] | Weak | 46.15% | 99.52% | Virtex 4 | 128 | $16 \times 64$ array |
| PicoPUF [26] | Weak | 45.60% | 98.74% | Artix-7 | 128 | 128 slices |
| RO PUF [6] | Weak | 48.05% | 99.30% | Artix-7 | 128 | >256 slices |
| Arbiter PUF [13] | Strong | 9.42% | 99.5% | Artix-7 | 64 | $129 \times 64$ slices |
| Flip-Flop Arbiter PUF [12] | Strong * | 40% | 97.10% | Artix-7 | 64 | $44 \times 64$ slices |
| p-SPUF [14] | Strong * | 50.58% | 93.3% | Artix-7 | 32 | $12 \times 32$ slices |
| L-PUF [15] | Strong * | 50.25% † | 96.3% ‡ | Artix-7 | 64 | 210 slices |
| LR-PUF [15] | Strong * | 49.66% † | 96.3% ‡ | Artix-7 | 64 | 129 slices |
| RO PUF (this work) | Strong * | 37.02% | 96.69% | Artix-7 | 64 | 174 slices |

## 6. Security Analysis

Physical Random Function, as a novel method of embedding a secret key onto the IC, is the answer to the conventional threat model where the adversary fabricates a "counterfeit" chip [2]. Although the application of PUFs significantly improves the security of IoT solutions, it still faces a variety of security threats. One of the most powerful ones is a modeling attack, where the attacker builds a software model of the PUF and intentionally collects a large set of CRPs to train the model [3].

Although our proposed design delivers a small hardware overhead with satisfactory performance, the resilience to modeling attacks should be further investigated and

discussed. The presented solution prevents the most straightforward attempt attack on RO-based PUF via a simple read out of a response bit for all pairs of oscillators, which would allow us to predict all responses with 100% correctness without knowing the exact RO frequencies themselves. However, it can still be attacked using machine learning algorithms, though the application of LFSR as a challenge generator may obfuscate the relation between the challenge and the response, making the application of machine learning attacks more challenging.

## 7. Conclusions

The paper presents a new lightweight PUF with a simple structure that can be conveniently implemented in FPGA. The proposed solution can be used as a Weak PUF for key generation purposes or a Strong PUF for device authentication. Compared to traditional Strong PUF, it has a significantly reduced complexity, resulting in low resource utilization and power consumption. These features are very suitable for applications in IoT resource-constrained devices. The PUF design was implemented on evaluation boards with Zynq 7000 devices equipped with 28 nm programmable logic. The experimental results show that it is satisfactory in uniformity, uniqueness, and reliability.

Future work involves investigating the proposed PUF's performance by using more boards and chips. It is also planned to conduct experiments by changing the environmental conditions, e.g., ambient temperature, humidity, and core voltage of the FPGA. Another plan involves the security evaluation of the PUF against machine learning attacks.

The authors also intend to enhance the presented PUF with Controlled PUF [3,27] elements and make it a part of Hardware Root of the Trust IP core, where it can be used for cryptographic key storage and management purposes.

**Author Contributions:** Conceptualization, M.K., R.G., and M.R.; methodology, M.K. and R.G.; software, M.K. and R.G.; validation, M.K. and R.G.; formal analysis, M.K. and R.G.; investigation, R.G. and M.B.; resources, M.R.; data curation, R.G. and M.B.; writing—original draft preparation, M.K., R.G., and M.B.; writing—review and editing, M.K., R.G., and M.R.; visualization, M.K. and R.G.; supervision, M.R. All authors have read and agreed to the published version of the manuscript.

**Funding:** This research was funded by The Polish National Centre for Research and Development under project No. CYBERSECIDENT/456446/III/NCBR/2020.

**Data Availability Statement:** Data is contained within the article.

**Conflicts of Interest:** The authors declare no conflicts of interest. The funders had no role in the design of the study; in the collection, analyses, or interpretation of the data; in the writing of the manuscript; or in the decision to publish the results.

## Abbreviations

The following abbreviations are used in this manuscript:

| | |
|---|---|
| ASIC | Application-Specific Integrated Circuit |
| CAD | Computer-Aided Design |
| CLB | Configurable Logic Block |
| CRP | Challenge–Response Pair |
| EEPROM | Electrically Erasable Programmable Read-Only Memory |
| FPGA | Field-Programmable Gate Array |
| IC | Integrated Circuit |
| IoT | Internet of Things |
| LFSR | Linear-Feedback Shift Register |
| LUT | Look-Up Table |
| NAND | Not-AND |
| NVRAM | Non-Volatile Random Access Memory |
| PUF | Physical Unclonable Function |
| RO | Ring Oscillator |

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
