# Peer review of "Lightweight Strong PUF for Resource-Constrained Devices"

_electronics, doi:10.3390/electronics13020351_

Round 1

Reviewer 1 Report

Comments and Suggestions for Authors

The submitted paper proposes a novel PUF architecture, which is implementable in IoT devices due to its resource efficiency. It extensively describes the architecture and the working mechanism too while also presenting experimental results obtained from physical device implementation. It is a well-written paper with only some minor language defects.

Remarks:
(C1) l47 The abbreviation PUF has been wrongly resolved into "Physical Random Functions"
(C2) l85 Statement "However, such an approach can be viewed as pseudo-random number generator enhanced with a PUF." needs more support. (Citation or discussion needed.)
(C3) l90 The showcase of the state-of-the-art is missing. (Potential competitors to the proposed method) Why is there yet another PUF approach needed? What are the shortcomings of the available ones? Are they not balancing well enough between resource and security? Why?
(C4) The related works section should describe the comparison bases [17-22] used for evaluation later.
(C5) Sec. 2.2 needs clearer description by showing how exactly a response is generated from a (given) challenge, maybe through an example. Also, consider using corresponding variables in Figure 1 and 2.
(C6) l152 Statement "However, this also will lead to a correlation of some of the response bits, especially if the initial value of LFSR is incorrectly selected." needs more support. (Citation or discussion needed.)
(C7) l154-160 It is unproved why this approach results in more uniqueness. It might obfuscate further the response generation alright, but it still has many rule-based (i.e. detectable) elements which may corrupt the uniqueness in the end. Try to validate your approach!
(C8) l177 Weak vs. Strong PUF characteristic of the system upon changing initial value supply approach needs proof or discussion at least.
(C9) Sec. 4 Define SLICE!
(C10) l215 Optimality ranges for the chip uniqueness (and maybe for reliability too) values should be defined too. (What is their target value?)
(C11) l234 It would be interesting to actually see how the choice of initial value influences the responses.
(C12) Sec 6 Conclusion needs expansion possibly by mentioning some future works / enhancement opportunities.

Comments on the Quality of English Language

Linguistic concerns:
- l107 by THE inputs called challenges
- l112 the PUF response equals TO the number
- l122 produced by THE PUF,
- l135 makes it unacceptable [to] FOR implementING
- l188 (somewhat) informal: It comes down to...

Reviewer 2 Report

Comments and Suggestions for Authors

In the abstract, it is not clear why this model is needed. Secondly, the authors are advised to add a couple sentences about their model followed by technology such IoT. At present, the abstract is very superficial.

It is strongly suggested the authors must summarize the contributions points in the introduction section. Please check IEEE Internet of Things, ACM or IEEE Transactions papers, how you can do this.

A flowchart diagram could be a good edition.

I want to see response of point 1 and 2 in the revised paper.

Please add a comment regarding the necessity of this work. The introduction and literature review currently lack a clear explanation for why this research is needed. Additionally, the author should highlight the limitations of each scheme considered in the literature.

Comments on the Quality of English Language

N/A

Reviewer 3 Report

Comments and Suggestions for Authors

1. [Introduction Section] The introduction provides an inadequate overview of the key contributions and novelty of the proposed approach. A more thorough comparison to prior art is needed to highlight what specifically is new in this work. At minimum, this section should span 3-4 paragraphs. 

2. [Introduction Section] The threat model considered in this work requires further elaboration. Ideally, one full paragraph should detail the specific types of attacks that the presented approach provides robustness against. Additionally, are there any threat scenarios that would compromise security? How does the threat model account for more sophisticated adversarial capabilities?

3. [Section 2.2] More background information is needed on existing ring oscillator PUFs in Section 2.2 to properly contextualize the proposed approach. This section could be expanded to 2-3 paragraphs providing greater technical detail on conventional designs and their limitations. Figures would also help illustrate the concepts.

4. [Section 3] The description of the scrambler module lacks critical implementation specifics. One paragraph should be added outlining the precise components and algorithms used in the scrambler design. This level of detail is crucial for reproducibility.

5. [Section 3] Using an LFSR-based scrambler seems to be a central design choice, yet the rationale is unclear. One to two sentences should justify selecting this type of scrambler over alternatives. What properties make an LFSR advantageous here?

6. [Section 3] The manuscript specifies using a generating polynomial for the LFSR scrambler but provides no further details. One to two sentences explaining the criteria for selecting the polynomial and any other relevant design choices would strengthen this section.

7. [Line 167] More information is essential regarding the buffer module implementation. At minimum, one paragraph should overview the key functions of this component and how it interacts with other system elements. Adding a block diagram figure could help illustrate this.

8. [Line 221] Choosing a 64-bit response length is stated but not properly motivated. One sentence providing the reasoning behind this parameter selection would improve the quality of presentation. What factors informed picking this particular value? 

9. [Line 225] Using only 1024 challenge values raises concerns over statistical significance. One sentence commenting on whether this sample size is sufficiently large compared to the overall challenge space could help justify the evaluation approach.

10. At minimum, one paragraph of security analysis is needed covering possible attack vectors against the system. This would greatly bolster claims around lightweight protection for IoT devices.

11. No analysis is provided on the scalability with increasing circuit area/complexity. One to two sentences discussing how the approach accommodates larger PUFs would be valuable. 

Round 2

Reviewer 1 Report

Comments and Suggestions for Authors

The paper duly improved.

Comments on the Quality of English Language

Only one small linguistic issue has been detected in the added text:

l112: may lead to [the] A suboptimal polynomial. If the polynomial is not A primitive one...

Reviewer 2 Report

Comments and Suggestions for Authors

Thanks for addressing my  previous round comment. At this point, I do not have further suggestions for the authors. 

Comments on the Quality of English Language

English is fine`